# Relationship between Kinesiophobia and Ankle Joint Position Sense and Postural Control in Individuals with Chronic Ankle Instability—A Cross-Sectional Study

**DOI:** 10.3390/ijerph19052792

**Published:** 2022-02-27

**Authors:** Mastour Saeed Alshahrani, Ravi Shankar Reddy

**Affiliations:** Department of Medical Rehabilitation Sciences, College of Applied Medical Sciences, King Khalid University, Abha 61421, Saudi Arabia; msdalshahrani@kku.edu.sa

**Keywords:** kinesiophobia, position sense, functional ankle instability, postural control

## Abstract

Functional ankle instability (FAI) is a condition that causes mechanical alterations to the ankle joint and leads to disability. Fear of movement can significantly influence physical factors, and understanding their relationship is crucial in assessing and managing individuals with FAI. The present study aimed to (1) assess the impact of kinesiophobia on ankle joint position sense (JPS) and postural control and (2) evaluate if kinesiophobia can predict JPS and postural control in FAI individuals. This cross-sectional study included 55 FAI individuals. The Tampa Scale of Kinesiophobia (TSK) score was used to measure kinesiophobia. The ankle JPS was evaluated using a digital inclinometer. The individuals were asked to actively reposition to the target position of 10° and 15° of dorsiflexion and plantarflexion. The reposition accuracy is measured in degrees. The static postural control was evaluated in unilateral stance using a stabilometric force platform, including assessments for the ellipse area, anterior to posterior sway, and medial to lateral sway in mm^2^. Kinesiophobia showed a significant positive correlation (moderate) with the ankle JPS errors in dorsiflexion (10°: r = 0.51, *p* < 0.001; at 15°: = r = 0.52, *p* < 0.001) and plantarflexion (10°: r = 0.35, *p* = 0.009; at 15°: = r = 0.37, *p* = 0.005). Kinesiophobia also showed significant positive (moderate) correlation with postural control variables (ellipse area: r = 0.44, *p* = 0.001; Anterior–Posterior sway: r = 0.32, *p* = 0.015; Medial–Lateral sway: r = 0.60, *p* < 0.001). Kinesiophobia significantly predicted ankle JPS (*p* < 0.05) and postural control (*p* < 0.05). Increased fear of movement is associated with increased ankle JPS errors and postural sway in FAI individuals. Therefore, assessment of these factors is critical in FAI individuals.

## 1. Introduction

The mechanical sciences view the human body as a sophisticated biomechanical system with kinematics and dynamics [1,2]. The key joints, which include the knee, shoulder, elbow, wrist, hip, ankle, cervical vertebrae, and lumbar vertebrae, are self-lubricating and practically frictionless, able to tolerate stress, torsion, and compression while still performing smooth and accurate movements [1,3]. The ankle joint is a congruent synovial joint with a single oblique axis that transfers weight to the proximal joints and allows smooth movements during functional activities [4]. Lateral ankle sprain is the most prevalent lower extremity musculoskeletal injury in physically active individuals [5]. Each year, an estimated 3 million individuals with ankle sprains receive standard care in a hospital emergency department [6]. Individuals with ankle sprains tend to have recurrences, and around 40% of people develop chronic symptoms such as pain, swelling, and instability leading to functional disability [7]. Previous studies have demonstrated that a majority of individuals with a repeated ankle sprain and persistent symptoms had functional ankle instability [8].

Individuals with chronic ankle instability demonstrate significant joint instability, a sense of giving way, and a lack of ankle motor control leading to functional disability [9]. The term “ chronic ankle instability “ refers to a condition that includes both components of functional and mechanical ankle instability [10]. Functional ankle instability (FAI) has been found in 50 percent of individuals with ankle injuries, with symptoms such as giving way, repeated sprains, and instability [8]. FAI can be caused and maintained by decreased muscle strength, altered proprioception, decreased postural control, delayed reflex actions, arthrogenic muscle inhibition, and subtalar instability [11,12,13].

Psychological variables play a role in developing chronic health disorders that impact proprioceptive function, muscle strength, and functional ability [14,15,16]. Kinesiophobia, also called “fear of movement or activity,” is defined as an excessive fear of physical movement, expecting or feeling of vulnerability to painful injury [17]. A reciprocal process occurs, resulting in a vicious cycle of negative thoughts and experiences, resulting in fear of movement, kinesiophobia, and catastrophic behavior [18,19,20,21]. Fear of movement may contribute to decreased muscular strength, increased postural sway, and impaired proprioception in musculoskeletal disorders [22,23,24]. Kinesiophobia can significantly impact ankle JPS and postural control, and these factors should be considered while rehabilitating patients with FAI to achieve a favorable and positive outcome [25,26].

The impact of kinesiophobia in individuals with FAI has received much attention. In this context, a substantial quantity of research has established a relationship between kinesiophobia, pain intensity, functional disability, and quality of life [27,28,29]. Increased baseline kinesiophobia scores were significantly associated with increased pain severity, functional impairment, and quality of life [27,28]. However, information is lacking addressing the relevance and direction of the findings in terms of ankle joint position sense (JPS) and postural control. This study was initiated to understand better and clarify the relationships between these factors and kinesiophobia. Examining kinesiophobia as a predictive and prognostic value on these outcomes would also help us better understand chronic musculoskeletal pain processes in FAI and improve clinical decision-making. Therefore, this study’s main goal was to (1) assess the relationship between kinesiophobia, JPS, and postural control and (2) assess if kinesiophobia can predict JPS and postural control in individuals with FAI.

## 2. Materials and Methods

### 2.1. Design 

After approval from the King Khalid University Institutional Research Review Committee (ECM #2021-6010), a cross-sectional study was undertaken at a medical rehabilitation department from March to December 2021. G*Power 3.1.9.4 statistical software (NeuIsenburg, Aichach, Germany) estimated the sample as 55, with α value of 0.05, β value of 0.2, and r of 0.5.

### 2.2. Participants

Fifty-five individuals with unilateral FAI were referred to medical rehabilitation clinics from orthopedic or general physicians. Licensed physical therapists, with a minimum of 10 years of experience in musculoskeletal physical therapy, conducted all evaluations. The subjects were included if they had (1) one or more lateral ankle sprains in the last six months, (2) history of giving way or feeling of ankle instability, (3) an Identification of Functional Ankle Instability score of ≥“11” [30]. The participants were excluded if they (1) had any lower extremity neuro-musculoskeletal abnormalities except ankle sprain that would limit testing, (2) had a history of previous fracture or dislocation or any surgeries of the lower extremities, (3) presented with cardiorespiratory or neurological disorders. 

The study followed the World Medical Association principles of the Declaration of Helsinki. All the individuals who voluntarily participated in this study were briefed on the study’s purpose, and all subjects in this study provided informed consent. The data collection took place in a single session.

### 2.3. Assessment of Kinesiophobia

The fear of movement or kinesiophobia was evaluated using the Tampa Scale for Kinesiophobia (TSK), a 17-item test [17]. Each test item was graded on a scale of 1 to 4, with 1 = strong disagreement and 4 = strong agreement. The total score ranges from 17 to 68, with 17 being the lowest and 68 being the highest. A TSK score of 37 or above indicates the presence of kinesiophobia [17]. TSK scale is a reliable (r = 0.78) and valid tool to measure fear of movement and has good internal consistency (Cronbach alpha = 0.80) [17,31].

### 2.4. Assessment of Ankle Joint Position Sense

The active to active ankle repositioning technique was considered to measure ankle JPS using a dual digital inclinometer (Figure 1). 

Individuals were requested to relocate their feet in dorsiflexion and plantarflexion directions to two targets (10° and 15°). During the JPS measurements, all patients were blindfolded to eliminate the contribution of vision. Subjects were instructed to sit on a couch with their eyes closed in a high sitting position. A velcro strap was used to secure one portion of the dual inclinometer to the lateral aspect of the tibia (mid-shaft). The primary inclinometer was fastened to the lateral border of the foot. The examiner guided the participant’s foot to the target angle of dorsiflexion or plantar flexion and was maintained for 5 s (asked to remember this position). Then, the foot was guided to a neutral or starting position. The participant was then instructed to reposition their foot to the target angle actively. The reposition accuracy is measured in degrees once the individual reaches the target position by saying “YES”. Three consecutive trials were performed in each tested direction, and the average of these three trials was considered for analysis. 

### 2.5. Assessment of Postural Control

Postural stability was assessed with a stabilometric force platform (IsoFree medical equipment, Tecnobody S.R.L., 2015). It consists of four primary components: (1) a stabilometric posture, (2) a 3D camera, (3) a touch screen, and (4) specialized software, all of which work together to analyze movement and postural control while providing real-time feedback (Figure 2).

The projection of the center of pressure (CoP) oscillations on the base of support as a result of the postural control process determines the ellipse surface area. The CoP is directly provided by the stabilometric force platform. To start with the postural control assessment, the device was calibrated, and the individuals were asked to stand with the single-leg stance on the force platform barefoot in a standardized position. Next, the participants were asked to flex the unsupported leg to keep it away from the force platform. Finally, the individuals were asked to look straight into the target mark in the computer monitor and maintain their postural stability with their arms resting by their sides. In each test, the individual was expected to stand on the testing leg for a period of 30 s, two trials were performed in each leg, and the best score was taken for analysis. The ellipse area, anterior to posterior, and medial to lateral sway were measured in mm^2^. 

### 2.6. Statistical Analysis

Data were analyzed using the SPSS version 24.0 (IBM Corporation, Armonk, NY, USA). The demographic characteristics of the study population were described using descriptive statistics. The study data were dispersed normally and tested with the Shapiro–Wilk test. The Pearson’s correlation coefficient (r) was estimated to determine the linear association between the kinesiophobia, ankle JPS, and postural control. Correlation coefficients between 0 and 0.3 were considered weak, 0.4 to 0.6 were considered moderate, and 0.7 to 1.0 were considered strong. A *p*-value of ≤0.05 was used to determine statistical significance. Simple linear regression analysis was used to assess if kinesiophobia can predict ankle JPS and postural control in FAI individuals. 

## 3. Results

A total of 55 individuals with a diagnosis of FAI were included in the study. The baseline physical and demographic characteristics of the study population are displayed in Table 1. 

There were significant differences in baseline ankle JPS and postural control variables between unaffected and normal leg (*p* < 0.001). The ankle JPS errors in dorsiflexion and plantarflexion were greater in the affected leg than the normal leg (Table 1). Additionally, the postural stability during the ellipse area, A–P sway, and M–L sway during single-leg stance were increased in the affected limb of FAI individuals (Table 1). 

TSK scores and their relationship with ankle JPS and postural control are summarized in Table 2 and Figure 3 and Figure 4. 

Kinesiophobia showed a significant positive correlation (moderate) with the ankle JPS in dorsiflexion (10°: r = 0.51, *p* < 0.001; at 15°: = r = 0.52, *p* < 0.001) and plantarflexion (10°: r = 0.35, *p* = 0.009; at 15°: = r = 0.37, *p* = 0.005). Kinesiophobia also showed significant positive (moderate) correlation with postural control variables (ellipse area: r = 0.44, *p* = 0.001; Anterior–Posterior sway: r = 0.32, *p* = 0.015; Medial–Lateral sway: r = 0.60, *p* < 0.001).

The linear regression analysis showed that kinesiophobia significantly predicted ankle JPS in dorsiflexion (10°: B = 0.24, *p* < 0.001; 15°: B = 0.26, *p* < 0.001) and plantarflexion (10°: B = 0.17, *p* = 0.009; 15°: B = 0.15, *p* = 0.005). Additionally, kinesiophobia significantly predicted the postural control ellipse area: B = 24.86, *p* = 0.001; Anterior–Posterior sway: B = 0.34, *p* = 0.015; Medial–Lateral sway: B = 0.42, *p* < 0.001) in individuals with FAI (Table 3).

## 4. Discussion

This study aimed to assess the relationship between kinesiophobia, JPS, and postural control and evaluate if kinesiophobia can predict JPS and postural control in individuals with FAI. This research revealed the following significant findings: Kinesiophobia showed a significant positive correlation with ankle JPS and postural control. In addition, kinesiophobia significantly predicted JPS and postural control in individuals with FAI.

In this study, individuals with FAI showed a mean TSK score of 41.36 (SD = 2.91). Lentz et al. [32] conducted a cross-sectional study with 85 patients with foot and ankle pathology to determine if kinesiophobia can impact foot and ankle mobility—factors such as decreased range of motion (ROM), age, and chronicity of ankle symptoms correlated with fear of movement. Pain-induced fear of movement was a single and significant contributor to the development of impairment and disability [32]. This dread of re-injury causes an individual to engage in avoidance conduct in order to reduce adverse consequences or prevent movement-related injury [32].

Kinesiophobia showed a significant moderate correlation with ankle JPS and postural control. This suggests that proprioceptive and motor control behavior is linked to a fear of movement. The most straightforward consequence from this study could be that fear of movement affects the ankle muscles, which is considered the most crucial source for afferent proprioceptive input from the ankle, operating irrespective of its structure force-generating capabilities [33]. A vicious cycle of maladaptive perceptions causes immobility and worsens pain perception, resulting in muscle atrophy, fibrosis, and functional impairment [34,35]. While FAI individuals perform ankle movements, kinesiophobia can activate altered motor patterns of the ankle muscles, impairing ankle JPS and postural control.

There are limited studies that evaluated these correlations in individuals with FAI. Asiri et al. [14] showed a significant positive correlation between kinesiophobia and cervical joint position errors in neck extension (r = 0.48, *p* < 0.001), right (r = 0.28, *p* < 0.025), and left (r = 0.31, *p* < 0.011) rotations [14]. In addition, kinesiophobia was found to be a significant predictor of neck J.P.S. [14]. The results of Alshahrani et al. [36] are in accordance with our study, which showed a significant positive correlation between kinesiophobia and knee JPS errors (*p* < 0.05) in individuals with bilateral knee osteoarthritis. Additionally, kinesiophobia significantly predicted knee proprioception (B = 0.96, *p* < 0.001) [36]. With these findings, we can see that kinesiophobia is an important component of the recovery process that should be considered when developing and administering rehabilitation programs for individuals with FAI.

Contrary to our study results, Aydogdu et al.’s [37] study did not demonstrate any relationship between kinesiophobia and knee JPS in anterior cruciate ligament reconstruction individuals. It is possible that the participants in the Aydogdu et al. study had lower kinesiophobia scores than the participants in this study. The respondents in our study had a mean kinesiophobia score of 41.36, whereas the subjects in the Aydogdu et al. study had a mean score of 36.54. In addition, we assessed ankle JPS at 10 and 15 degrees of plantar flexion and dorsiflexion, respectively, because these angles are more functional. Perhaps the results would have been different if we had chosen alternative angles or dorsiflexion or plantarflexion end ranges.

The second finding in this study, kinesiophobia, showed a positive relationship with postural control. Masood et al. [24] showed a significant positive correlation between fear of movement and increased Anterior–Posterior sway and Medial–Lateral sway in subjects with chronic low back pain [24]. This study population with FAI leg also showed a larger ellipse area, Anterior–Posterior, and the Medial–Lateral sway compared to the asymptomatic leg. Patients with FAI continue to have impaired postural control, even when not experiencing pain. It is possible that the appearance of these changes in the absence of an active pain episode is due to kinesiophobia [38]. According to the fear-avoidance model, viewing pain as catastrophic can lead to fear of pain and avoidance of actions that are thought to aggravate pain or induce re-injury [38,39]. The trend for patients with FAI to sway was more consistent with our hypothesis that the fear of movement impacts postural sway.

This study’s findings have clinical relevance for individuals with FAI undergoing rehabilitation. Because the observed changes in postural sway in FAI were mostly mediated by fear of movement, kinesiophobia management should be addressed as a therapeutic component of the rehabilitation process to reestablish adequate postural control.

### Limitations

The study’s strength and capacity to extrapolate the findings to a larger population are limited by its small sample size. There was no healthy control group included in the study to assess the level of kinesiophobia between FAI and healthy individuals. We could only draw inferences at a particular time point because we used a cross-sectional study design. We did not conduct multivariate analysis because it would have allowed us to account for confounding variables such as age and time since the last injury, which could have skewed the results of this correlations study. Future research should focus on the duration of ankle sprains and their associated effect on proprioception and postural control. In addition, investigations aimed at prevention and interventional strategies for kinesiophobia, ankle JPS, and postural control should be explored in FAI individuals.

## 5. Conclusions

This study shows that kinesiophobia is significantly correlated to ankle JPS and postural control. Additionally, kinesiophobia significantly predicted JPS and postural control in FAI individuals. These factors should be incorporated into the diagnostic process to ensure a holistic and multi-modal approach to FAI individuals’ comprehension assessment, planning, and rehabilitation.

## Figures and Tables

**Figure 1 ijerph-19-02792-f001:**
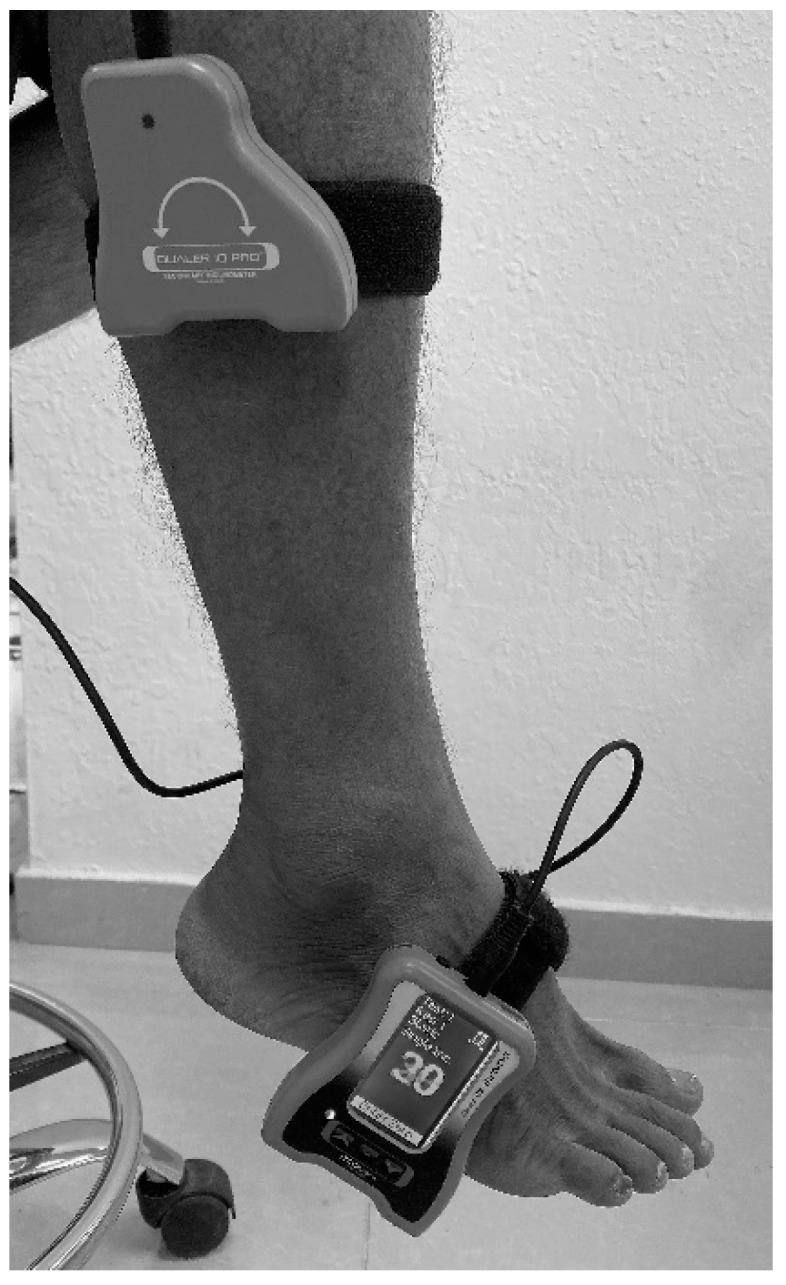
Assessment of ankle joint position sense using a dual digital inclinometer.

**Figure 2 ijerph-19-02792-f002:**
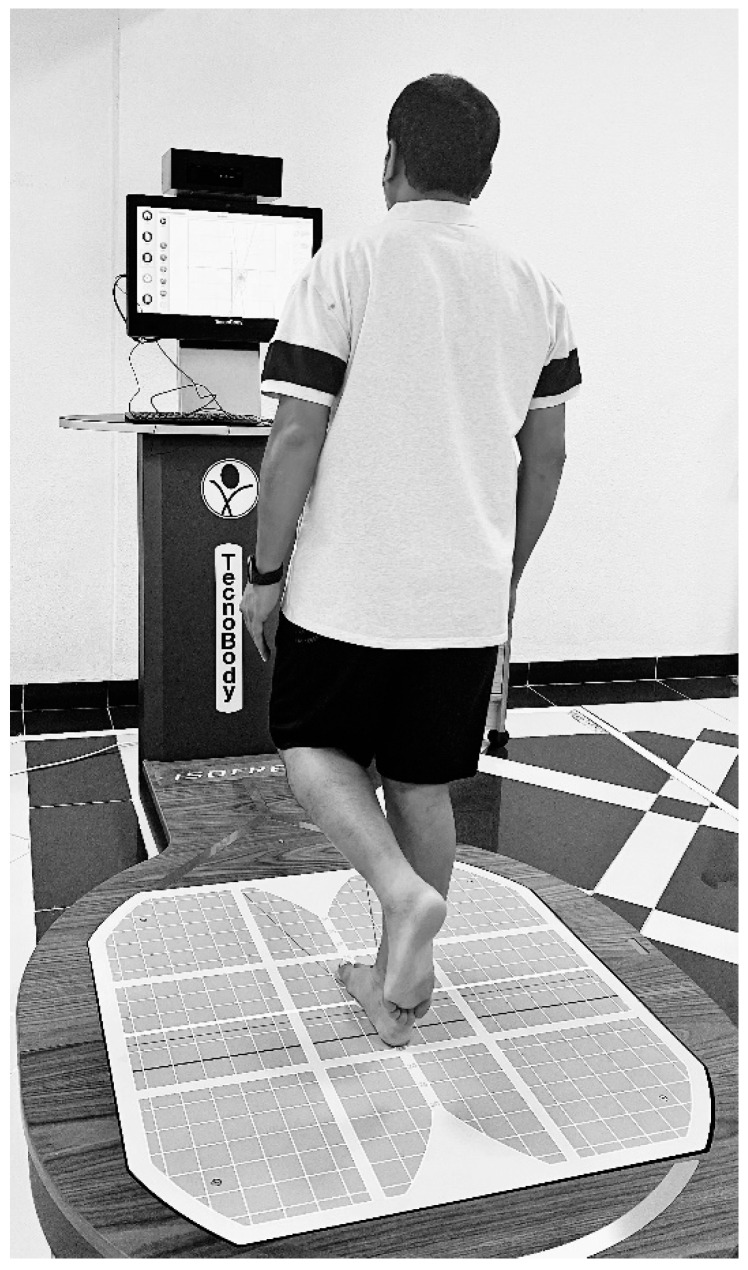
Assessment of postural control using stabilometric force platform.

**Figure 3 ijerph-19-02792-f003:**
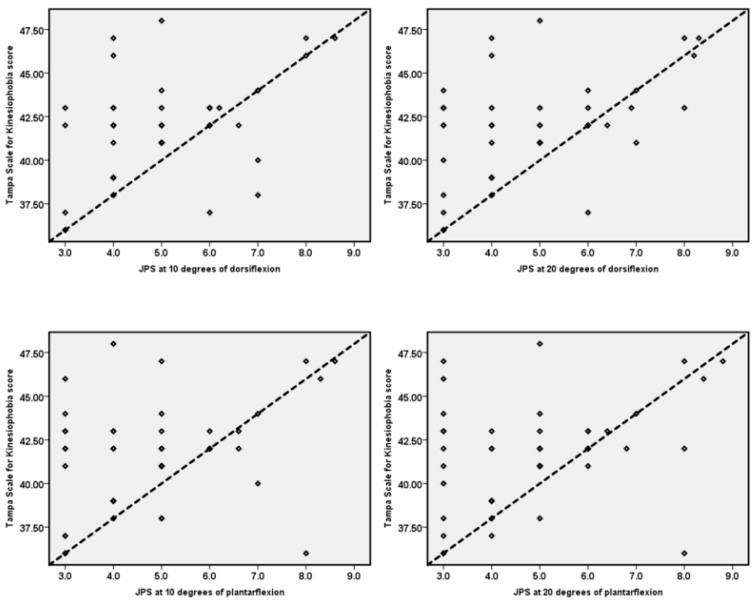
Relationship between TSK scores and ankle joint position sense.

**Figure 4 ijerph-19-02792-f004:**
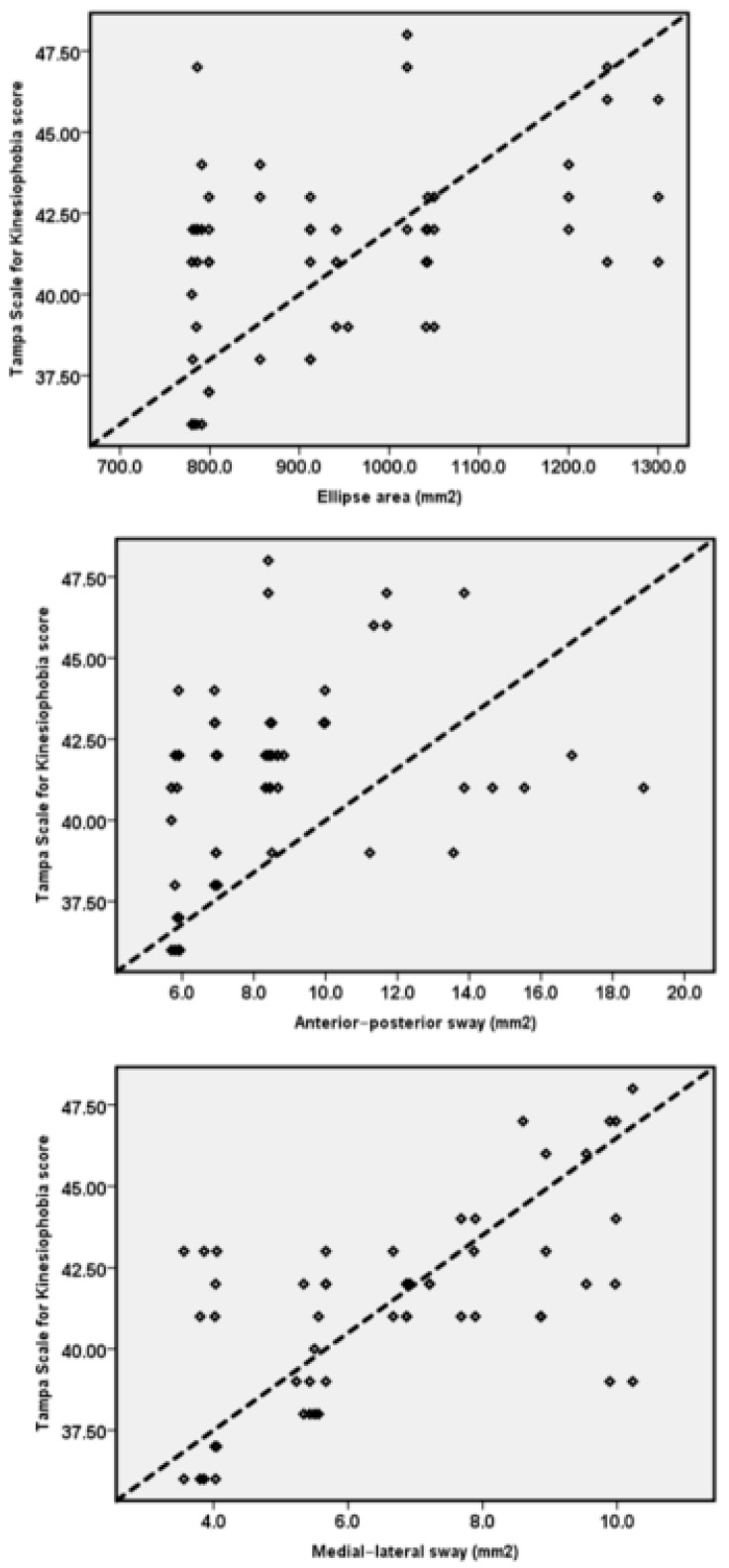
Relationship between TSK scores and postural control variables.

**Table 1 ijerph-19-02792-t001:** Physical and demographic characteristics of participants (*n* = 55).

Variables	FAI Group Mean ± SD	*p*-Value
Age (years)	23.00 ± 1.91	-
Gender: M: F (%)	34 (42.5):21 (26.3)	-
BMI (kg/m^2^)	24.41 ± 2.67	-
Duration of Ankle Sprain (Months)	12.47 ± 3.84	-
TSK sore	41.36 ± 2.91	-
JPS at 10 degrees of dorsiflexion Affected Normal	4.93 ± 1.38 1.35 ± 1.04	<0.001
JPS at 15 degrees of dorsiflexion Affected Normal	4.80 ± 1.50 1.33 ± 1.16	<0.001
JPS at 10 degrees of plantarflexion Affected Normal	4.82 ± 1.49 1.11 ± 0.79	<0.001
JPS at 15 degrees of plantarflexion Affected Normal	4.84 ± 1.56 1.33 ± 0.70	<0.001
Ellipse area (mm^2^) Affected Normal	946.75 ± 166.13 462.22 ± 155.53	<0.001
Anterior–Posterior sway (mm^2^) Affected Normal	8.73 ± 3.08 3.67 ± 1.40	<0.001
Medial–Lateral sway (mm^2^) Affected Normal	6.67 ± 2.10 3.57 ± 1.57	<0.001

BMI = Body mass index, TSK = Tampa Scale for Kinesiophobia, JPS = joint position sense.

**Table 2 ijerph-19-02792-t002:** Coefficient of correlation between TSK scores, joint position sense, and postural control (*n* = 55).

Explanatory Variables	TSK Scores	
	r	*p*-Value
JPS at 10 degrees of dorsiflexion	0.51	<0.001
JPS at 15 degrees of dorsiflexion	0.52	<0.001
JPS at 10 degrees of plantarflexion	0.35	0.009
JPS at 15 degrees of plantarflexion	0.37	0.005
Ellipse area (mm^2^)	0.44	0.001
Anterior–Posterior sway (mm^2^)	0.32	0.015
Medial–Lateral sway (mm^2^)	0.60	<0.001

TSK = Tampa Scale for Kinesiophobia, JPS = joint position sense.

**Table 3 ijerph-19-02792-t003:** General linear regression of TSK and explanatory variables (*n* = 55).

Explanatory Variables	B	SE.	*t*-Value	*p* Value
JPS at 10 degrees of dorsiflexion	0.24	0.05	4.28	<0.001
JPS at 15 degrees of dorsiflexion	0.26	0.06	4.41	<0.001
JPS at 10 degrees of plantarflexion	0.17	0.06	2.71	0.009
JPS at 15 degrees of plantarflexion	0.20	0.06	2.93	0.005
Ellipse area (mm^2^)	24.86	7.06	3.51	0.001
Anterior–Posterior sway (mm^2^)	0.34	0.13	2.50	0.015
Medial–Lateral sway (mm^2^)	0.42	0.08	5.39	<0.001

JPS = joint position sense; B = Unstandardized Coefficients, SE = standard error.

## Data Availability

The data presented in this study are available on request from the corresponding author.

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
