# Peer review of "Relationship between Kinesiophobia and Ankle Joint Position Sense and Postural Control in Individuals with Chronic Ankle Instability—A Cross-Sectional Study"

_ijerph, 2022, doi:10.3390/ijerph19052792_

Round 1
Reviewer 1 Report
The objective of this study was to study the influence between kinesiophobia, ankle joint position sense and postural control. The feasibility of using motor phobia to predict JPS and posture control was also evaluated. This study holds merit and the authors deserve recognition for their efforts in conducting this study.
However, the authors should address some concerns before the final decision.
1.From the perspective of mechanical sciences, the human body is a complex biomechanical system, including kinematics and dynamics. The important joints, including the knee, shoulder, elbow, wrist, hip, ankle, cervical vertebra and lumbar vertebrae, are self-lubricating, almost frictionless, and are able to withstand tension, torsion and compression and maintain heavy loads while still executing smooth and precise movements. Please emphasize the importance of the ankle joint in all joints.
- In section “2. Materials and Methods”, please provide basic theory of research, especially mathematical method, such as behavior change wheel, BCW.
- In 2.5. Assessment of Postural control, please provide information about the center of pressure (CoP). Is it directly given by the stabilometric force platform?
- There are sentences with repeated references in the article. For example, ‘Contrary to our study results, Aydogdu et al.'s [32] study did not demonstrate any 249 relationship between kinesiophobia and knee JPS in anterior cruciate ligament reconstruc-250 tion individuals [32].’
- Please modify some unit formats in the article. For example, mm2 should be modified into mm2.
- There is ‘#null!’ in the submitted Excel file.
- In figure 3, the font size of the axis title is a little small.
- Most publications focus on the athletic injury, mechanical kinesthetics, biodynamics, biomechanical properties of the joint, as well as the underlying medical biochemical mechanism. Please optimize your references with regards to Biomechanical performance design of joint prosthesis for medical rehabilitation, Biomechanical Strengthening for Limb Articulation, etc.
Author Response
Response to Reviewer comments
Thank you for your effort and time in reviewing our manuscript. The reviewing process has significantly improved the quality of this manuscript. Therefore, I am submitting this "Response to reviewers" document summarizing the changes we made in response to the critiques. I have highlighted the changes in the manuscript and provided the line numbers where the changes have been made.
|
Reviewer 1 |
|
||
|
Sl.no |
Queries |
Response to queries |
Changes made in the manuscript - Line Numbers |
|
1 |
1.From the perspective of mechanical sciences, the human body is a complex biomechanical system, including kinematics and dynamics. The important joints, including the knee, shoulder, elbow, wrist, hip, ankle, cervical vertebra, and lumbar vertebrae, are self-lubricating, almost frictionless, and can withstand tension, torsion and compression and maintain heavy loads while still executing smooth and precise movements. Please emphasize the importance of the ankle joint in all joints. |
· The introduction is modified. · WE emphasized the importance of ankle joints in all joints. · The suggested changes are implemented in the manuscript. |
32 to 38 |
|
2 |
In section “2. Materials and Methods”, please provide basic theory of research, especially mathematical method, such as behavior change wheel, BCW. |
· We have not considered the behavior change wheel (BCW) to provide the basic theory of research. · We have five steps approach for BCW. o 1. Identify the problem o 2. Review the evidence o 3. Draw a logic model o 4. Identify indictors and collect monitoring data o 5. Evaluate logic model​ · Steps 1-3 should be carried out before the project begins · Step 4 (monitoring) should continue from the very start to the end of your project (and, ideally, beyond). · Step 5 (analysis) should not be left to the end either. Interim and ongoing evaluations will enable you to improve your project or service. |
N/A |
|
|
In 2.5. Assessment of Postural control, please provide information about the center of pressure (CoP). Is it directly given by the stabilometric force platform? |
· Yes, sir, The CoP is directly given by the stabilometric force platform. · The sentence is mentioned in the methods. |
138-139 |
|
|
There are sentences with repeated references in the article. For example, ‘Contrary to our study results, Aydogdu et al.'s [32] study did not demonstrate any 249 relationship between kinesiophobia and knee JPS in anterior cruciate ligament reconstruc-250 tion individuals [32].’ |
· The repeated references are now deleted. |
238-243 |
|
|
Please modify some unit formats in the article. For example, mm2 should be modified into mm2. |
· The “mm2” is modified into “mm2” |
21 147 |
|
|
There is ‘#null!’ in the submitted Excel file. |
· The excel file is now corrected, removing “null”. |
N/A |
|
|
In figure 3, the font size of the axis title is a little small. |
· The figures are uploaded in supplementary files. · The axis titles become much clear when they are magnified. · The figures are produced as per the standard publication guidelines. |
Figure 3 |
|
|
Most publications focus on the athletic injury, mechanical kinesthetics, biodynamics, biomechanical properties of the joint, as well as the underlying medical biochemical mechanism. Please optimize your references with regards to Biomechanical performance design of joint prosthesis for medical rehabilitation, Biomechanical Strengthening for Limb Articulation, etc. |
· The references are modified focusing on Biomechanical performance design of joint prosthesis for medical rehabilitation, Biomechanical Strengthening for Limb Articulation |
References |

Reviewer 2 Report
The authors present a well written study describing the associations between kinesiophobia and postural sense and control in patients with functional ankle instability. The methodology is sound and the authors cleverly chose patients with unilateral pathology who served as their own controls by analyzing the unaffected side; which is a strength of their study design. Points to clarify are described below.
Major comments:
- Mechanical joint instability factors as part of the acute injury/sprain may potentiate the effects of kinesiophobia on proprioception and postural control, and this may have confounded the results in patients with a more recent sprain vs patients who recovered weeks from an acute injury prior to the study. Is this what duration of ankle sprain in Table 1 refers to? Otherwise, please report data on ankle sprain timing and whether the patients were symptomatic from it at the time of the study, and define ankle sprain duration. Please discuss the possible effect of sprain duration and timing on proprioception and postural control.
- Ideally, the associations between kinesiophobia and proprioception/postural control should be adjusted by time of most recent injury, although small sample size may be a limiting factor.
- The authors should include as part of their limitations the fact that a multivariate review was not conducted, as this would have allowed to adjust for factors such as age and last injury time, which may be confounding the correlations.
- The authors should remove recall bias as part of their limitations, as the definition of recall bias does not apply to this study design.
Minor comments:
- The mean age listed in page 2, line 78 is different from that listed in Table 1. Which one is accurate? The authors should leave this information in the results and remove it from the methods.
- Please report percentage of male and female participants as part of demographics
- In pages 1-2, lines 43-45: do the authors refer to 50% of individuals with ankle injuries?
- In Pages 1-2, lines 43-45: giving way is listed twice
- Page 5 line 167-168: All individuals included in the study had FAI. Please rephrase statement about postural stability to say that the measures were increased in the affected extremity, not in individuals with FAI.
- Page 4, line 153 page 4: Please rephrase to indicate p value used to determine statistical significance rather than percentage
- Please define ROM at first use
Author Response
Response to Reviewer comments
Thank you for your effort and time in reviewing our manuscript. The reviewing process has significantly improved the quality of this manuscript. Therefore, I am submitting this "Response to reviewers" document summarizing the changes we made in response to the critiques. I have highlighted the changes in the manuscript and provided the line numbers where the changes have been made.
|
Reviewer 2 |
|||
|
Sl.no |
Queries |
Response to queries |
Changes made in the manuscript - Line Numbers |
|
1. |
Mechanical joint instability factors as part of the acute injury/sprain may potentiate the effects of kinesiophobia on proprioception and postural control, and this may have confounded the results in patients with a more recent sprain vs patients who recovered weeks from an acute injury prior to the study. Is this what duration of ankle sprain in Table 1 refers to? Otherwise, please report data on ankle sprain timing and whether the patients were symptomatic from it at the time of the study and define ankle sprain duration. Please discuss the possible effect of sprain duration and timing on proprioception and postural control. Ideally, the associations between kinesiophobia and proprioception/postural control should be adjusted by time of most recent injury, although small sample size may be a limiting factor. |
The participants' mean duration of ankle sprain was 12.47 ± 3.84 months, as presented in table 1. I agree with your comment that patients with a more recent sprain may have greater kinesiophobia. The subjects were included if they had 1) one or more lateral ankle sprains in the last six months, 2) history of giving way or feeling of ankle instability, 3) an Identification of Functional Ankle Instability score of ≥ "11". I concur with you that this study had a limited sample size, that controlling for the time since the most recent injury is challenging, and that looking for associations is difficult. |
265-274 |
|
2. |
The authors should include as part of their limitations the fact that a multivariate review was not conducted, as this would have allowed to adjust for factors such as age and last injury time, which may be confounding the correlations. |
The limitations are modified. The suggested information is included as one of the limitations. |
265-274 |
|
3. |
The authors should remove recall bias as part of their limitations, as the definition of recall bias does not apply to this study design. |
The recall bias mentioned as part of the limitation is removed. |
265-274 |
|
4. |
The mean age listed in page 2, line 78 is different from that listed in Table 1. Which one is accurate? The authors should leave this information in the results and remove it from the methods. |
The mean age of the participants was 23.0 ± 1.91 years. The mean age is removed from the methods section. |
Table 1. |
|
5. |
Please report percentage of male and female participants as part of demographics |
The number and percentage of males and females are reported in table 1. |
Table 1 |
|
6. |
In pages 1-2, lines 43-45: do the authors refer to 50% of individuals with ankle injuries?
|
Sorry for the typo error. Yes, it refers to 50% of individuals with ankle injuries. The sentence is modified. |
48-50 |
|
7. |
In Pages 1-2, lines 43-45: giving way is listed twice
|
The repeated word “giving way” is deleted. |
49-50 |
|
8. |
Page 5 line 167-168: All individuals included in the study had FAI. Please rephrase statement about postural stability to say that the measures were increased in the affected extremity, not in individuals with FAI.
|
The sentence is modified. |
171 |
|
9. |
Page 4, line 153 page 4: Please rephrase to indicate p value used to determine statistical significance rather than percentage
|
The sentence is modified. |
156- |
|
10. |
Please define ROM at first use |
The “ROM” is defined. |
212 |

Round 2
Reviewer 2 Report
Thank you for addressing the comments